# A multicenter, single-arm study using a modified faricimab treat-and-extend regimen in patients with macular edema due to central retinal vein occlusion: RVOSTAR study design protocol

Mineo Kondo[1]*, Masahiko Shimura[2], Akitaka Tsujikawa[3], Yuki Muraoka[3], Takahiro Kogo[3], Yuki Akiyama[3], Yuki Hama[3], Jun Tsujimura[4], Keisuke Iwasaki[4], Motohiro Kamei[5]

**1** Department of Ophthalmology, Mie University Graduate School of Medicine, Mie, Japan, **2** Department of Ophthalmology, Tokyo Medical University Hachioji Medical Center, Tokyo, Japan, **3** Department of Ophthalmology and Visual Sciences, Kyoto University Graduate School of Medicine, Kyoto, Japan, **4** Chugai Pharmaceutical Co., Ltd., Tokyo, Japan, **5** Department of Ophthalmology, Aichi Medical University, Aichi, Japan

* mineo@med.mie-u.ac.jp

## Abstract

### Introduction

Anti–vascular endothelial growth factor (anti-VEGF) agents are generally considered to be the first line of therapy for macular edema due to retinal vein occlusion (RVO-ME). However, current anti-VEGF treatment regimens in Japan are unable to maintain long-term vision improvement, particularly in patients with central RVO-ME (CRVO-ME). Faricimab is a dual angiopoietin-2/VEGF inhibitor approved for the treatment of RVO-ME in Japan. The RVOSTAR study is being conducted to evaluate the long-term maintenance of vision outcomes with faricimab using a modified treat-and-extend (T&E) regimen in treatment-naïve Japanese patients with CRVO-ME or hemi–RVO-ME (HRVO-ME) and to assess the factors affecting visual acuity and treatment intervals.

### Methods and design

RVOSTAR (Japan Registry of Clinical Trials; jRCTs041250001) is an unmasked, single-arm, multicenter, prospective interventional study. Patients with CRVO-ME or HRVO-ME will 1) receive faricimab 6 mg every 4 weeks (≤6 injections) until ME is resolved; 2) be observed with no treatment until ME reoccurs, based on pre-specified central subfield thickness (CST) criteria; and then 3) be treated according to a T&E regimen up to 72 weeks, with dosing intervals based on time to relapse and adjusted by 4-week increments (minimum interval: every 4 weeks; maximum interval: no limit). Vision outcomes include best-corrected visual acuity (BCVA) and CST. The primary

which permits unrestricted use, distribution, and reproduction in any medium, provided the original author and source are credited.

**Data availability statement:** No datasets were generated or analysed during the current study. All relevant data from this study will be made available upon study completion.

**Funding:** This study was funded by Chugai Pharmaceutical Co., Ltd., a member of the Roche Group. Chugai Pharmaceutical Co., Ltd. was involved in the study design and preparation of the manuscript. Medical writing assistance was provided by Nilisha Fernando, PhD, of Envision Pharma Group, and funded by Chugai Pharmaceutical Co., Ltd. Envision Pharma Group's services complied with international guidelines for Good Publication Practice (GPP 2022).

**Competing interests:** The authors of this manuscript have the following competing interests: M.Kondo: financial support from Alcon Japan, Chugai, HOYA, Santen, and Senju; consulting fees from Chugai; royalties from Alcon Japan, Bayer, HOYA, Kowa, Novartis, Otsuka, Santen, and Senju. M.S.: financial support from Bayer, Chugai, Kowa, Novartis, Otsuka, Roche, Santen, and Senju; consulting fees from Bayer, Boehringer Ingelheim, Chugai, Cimic, and Nikki HD. A.T.: financial support from Novartis, Santen, and Senju; consulting fees from Santen and Senju; renumeration from Bayer, Chugai, Novartis, Santen, and Senju. Y.M.: financial support from Alcon Japan, AMO Japan, Bayer Yakuhin, Canon, Chugai, HOYA, Johnson & Johnson K.K., Novartis, ROHTO, Santen, and Senju. T.K., Y.A., Y.H.: none. J.T., K.I.: employees of Chugai Pharmaceutical Co., Ltd. M.Kamei: financial support from Alcon Japan, Chugai, HOYA, Kowa, Menicon, Otsuka, Santen, Senju, Sony, and Wakamoto; royalties from Alcon Japan, Bayer, Chugai, Daicel, HOYA, Kowa, Novartis, Santen, Senju, Sony, and Wakamoto. Chugai Pharmaceutical Co., Ltd., the funder, has provided employment (J.T., K.I.), consulting fees (M.Kondo, M.S.), financial support (M.Kondo, M.S., Y.M., M.Kamei), renumeration (A.T.), and royalties (M.Kamei) to the authors. There are no patents, products in development, or marketed products associated with this research to declare. This does not alter our adherence to PLOS ONE policies on sharing data and materials.

endpoint is the change from baseline in BCVA at week 72. Safety outcomes include ocular and non-ocular adverse events.

## Conclusion

RVOSTAR will evaluate the long-term maintenance of vision outcomes with a modified faricimab T&E regimen in patients with CRVO-ME or HRVO-ME while reducing the burden associated with frequent injections. The findings from this study may help to optimize dosing frequency in clinical practice.

## Introduction

Retinal vein occlusions (RVOs) are a group of common retinal diseases that can impair retinal circulation and cause several complications, including cystoid macular edema (ME) and vision loss [1]. There are three types of RVO depending on the site of occlusion: branch RVO, hemi-RVO (HRVO), and central RVO (CRVO). CRVO arises from an occlusion within or posterior to the optic nerve head, which contains the central retinal vein. In Japan, the cumulative incidence of any RVO type was 3.0% and the cumulative incidence of CRVO was 0.3% between 1998 and 2007 [2].

Anti–vascular endothelial growth factor (anti-VEGF) agents are considered to be the first line of therapy for ME due to RVO (RVO-ME) [1]. However, current anti-VEGF treatment regimens are unable to maintain long-term vision improvement, particularly for patients with CRVO-ME [3–5]. For example, in the phase 3 CRUISE trial, anti-VEGF therapy with ranibizumab significantly improved visual acuity in patients with CRVO-ME during the first year of treatment [6]. However, in HORIZON, the open-label, single-arm, multicenter extension study of CRUISE, vision loss was observed at 2 years following pro re nata (as needed) treatment with an extended monitoring interval of 3 months, corresponding to a decrease in ranibizumab injections [3]. Similarly, in the phase 3 GALILEO and COPERNICUS trials, improvements in vision outcomes were observed with monthly aflibercept treatment until week 24, after treatment was switched to pro re nata dosing, as well as after week 52, when the monitoring interval was increased from monthly to every 8–12 weeks (Q8–12W; Q8W up to week 72 in GALILEO and Q12W up to week 100 in COPERNICUS). Of note, improvements in vision outcomes initially observed at week 24 had diminished by the final observation [4,5]. As monthly monitoring for prolonged periods is often too burdensome for patients and physicians, a personalized treat-and-extend (T&E) approach offers an alternative treatment strategy, in which dosing intervals are extended while proactively administering the drug according to each patient's disease activity (based on pre-specified vision and anatomic criteria), thereby reducing burden while maintaining vision outcomes [7].

Faricimab is a dual angiopoietin-2/VEGF inhibitor, which was approved in Japan (March 2024) for the treatment of RVO-ME at a dose of 6 mg (dosing interval ≥4 weeks) [8]. In the pivotal global phase 3 COMINO trial (NCT04740931) in patients with CRVO-ME or HRVO-ME, a personalized faricimab T&E regimen was well

tolerated, and improvements at week 24 (after Q4W dosing) were maintained through week 72 with subsequent T&E dosing (median ~4–5 injections between 24 and 72 weeks) [9–11]. The short-term efficacy of faricimab in treating RVO-ME has been demonstrated in recent real-world studies, including in patients who switched from prior anti-VEGF therapy to faricimab Q4W [12], as well as in patients who were treatment-naïve or those who switched treatment regimens to pro re nata faricimab [13]. However, in clinical practice in Japan, optimal T&E dosing has not been explored for RVO-ME. This is an important consideration given that frequent anti-VEGF injections can impose a large burden on patients and the healthcare system, with issues such as treatment adherence potentially impacting outcomes [14].

The aim of the RVOSTAR study is to investigate the long-term maintenance of vision outcomes with faricimab using a modified T&E regimen in treatment-naïve Japanese patients with CRVO-ME or HRVO-ME and to assess the factors affecting visual acuity and treatment intervals. This manuscript summarizes the RVOSTAR study design, including the patient eligibility criteria, treatment protocol, study assessments, and outcomes.

## Materials and methods

### Trial overview

The Japanese "**RVO** treatment **S**trategy with modified **T**reat **A**nd extend **R**egimen of faricimab" (RVOSTAR; Japan Registry of Clinical Trials; jRCTs041250001 [15]) study is an unmasked, single-arm, multicenter, prospective interventional clinical trial to investigate the efficacy and safety of a modified faricimab T&E dosing regimen in treatment-naïve patients with CRVO-ME or HRVO-ME. The study will be conducted in compliance with the Declaration of Helsinki (Japan Medical Association translation) [16], the Clinical Trials Act (Act No. 16 of 2017) [17], and the Enforcement Regulations of the Clinical Trials Act (Ministry of Health, Labour and Welfare Ordinance No. 17 of 2018) [17]. Written informed consent will be obtained by all participants. The RVOSTAR study was approved by the Mie University Hospital Certified Review Board during March 2025 [15].

### Participants and eligibility criteria

Eligible patients with CRVO-ME or HRVO-ME will be screened and enrolled into the study. Inclusion criteria and key exclusion criteria are detailed in Table 1. Briefly, eligible patients will be aged ≥18 years with recently (within 4 months) diagnosed foveal center–involved CRVO-ME or HRVO-ME in the study eye who are naïve to treatment of ME (including anti-VEGF treatment and steroids). Patients will be excluded if they have any current ocular condition that may cause vision loss other than CRVO-ME or HRVO-ME (including diabetic ME, and neovascular age-related macular degeneration); a history of idiopathic uveitis in either eye; systemic treatment for infection; or any periocular or intravitreal treatment for retinal diseases. The full list of exclusion criteria is outlined in S1 Table.

### Treatment protocol and assessments

A schedule of enrollment, visits, interventions, and assessments in RVOSTAR is presented in Fig 1. Patients will receive intravitreal injections of faricimab 6 mg initially Q4W until ME is resolved, and then following an observation period, patients will be treated according to a modified T&E regimen up to 72 weeks (Fig 2). In the *induction phase*, patients will receive injections on day 1 and then Q4W (up to 6 injections) until resolution of ME (defined as central subfield thickness [CST] <325 μm as measured on Spectralis spectral-domain optical coherence tomography [SD-OCT], or CST <315 μm as measured on Cirrus SD-OCT, Topcon SD-OCT, or an equivalent OCT). The CST value at the visit at which resolution of ME is confirmed will be the reference CST. Study visits will also occur Q4W from day 1 through week 24. In the *observation phase*, faricimab will not be administered. Study visits will occur Q4W from resolution of ME until disease activity is observed (CST increase of ≥20% from resolution of ME, or the lowest CST value if ME is not resolved after 6 injections). In the *maintenance phase*, patients will visit the study site at a frequency depending on the T&E dosing interval. The initial dosing interval during this phase will be

**Table 1. Eligibility criteria for screening and enrollment into the RVOSTAR study.**

| Inclusion criteria | Key exclusion criteria[a] |
|---|---|
| • Age ≥ 18 years and able to provide informed consent<br>• Participate in all scheduled visits and assessments<br>For the study eye:<br>• Foveal center–involved CRVO-ME or HRVO-ME[b], diagnosed within 4 months before the screening visit[c]<br>• Naïve to treatment of CRVO-ME or HRVO-ME (including IVT anti-VEGF injections and steroids)<br>• Decimal visual acuity of 0.5–0.05 as assessed on the visual acuity test on day 1 (pre-treatment)<br>• CST of ≥325 μm on Spectralis SD-OCT or ≥315 μm on Cirrus SD-OCT, Topcon SD-OCT, or other equivalent OCT<br>• Sufficiently clear ocular media and adequate pupillary dilation to allow acquisition of good quality retinal images to confirm diagnosis | • Systemic treatment for suspected or active systemic infection<br>• Stroke (cerebral vascular accident) or myocardial infarction within 6 months<br>• Uncontrolled blood pressure (defined as systolic >180 mmHg and/or diastolic >110 mmHg while a patient is at rest)<br>• Active cancer within 12 months[d]<br>• Significant disease, surgery, or systemic steroids within 1 month<br>For the study eye:<br>• History of retinal detachment or macular hole (stage 3 or 4)<br>• Cataract surgery or treatment for complications of cataract surgery within 3 months<br>• Any prior or current periocular or IVT treatment (including anti-VEGF) for ME, macular neovascularization, or vitreomacular-interface abnormalities<br>• Prior periocular or IVT treatment (including anti-VEGF) for other retinal diseases<br>• Macular laser (focal/grid) or PRP in the study eye, or PRP scheduled within 3 months<br>For both eyes:<br>• Any history of idiopathic or immune-mediated uveitis in either eye<br>• Active ocular inflammation or suspected or active ocular or periocular infection in either eye |

CRVO-ME, macular edema due to central retinal vein occlusion; CST, central subfield thickness; HRVO-ME, macular edema due to hemi-retinal vein occlusion; IVT, intravitreal; ME, macular edema; mmHg, millimeters of mercury; OCT, optical coherence tomography; PRP, panretinal photocoagulation; SD-OCT, spectral-domain optical coherence tomography; SS-OCT, swept-source optical coherence tomography; VEGF, vascular endothelial growth factor.

[a]Full exclusion criteria are presented in S1 Table.

[b]CRVO is defined as retinal hemorrhage, telangiectatic capillary bed, dilated venous system, or other biomicroscopic evidence of RVO (neovascularization or vitreous hemorrhage) in the entire retina and HRVO as those in 2 quadrants of the retina drained by the affected vein.

[c]Based on SD-OCT or SS-OCT images.

[d]Except for appropriately treated carcinoma in situ of the cervix, non-melanoma skin carcinoma, and prostate cancer with a Gleason score of ≤6 and a stable prostate-specific antigen for >12 months.

the same as the dosing interval during the observation phase. Dosing intervals will be adjusted by 4-week intervals (S1 Fig). If CST increases by <10% (from reference CST), the interval will be extended by 4 weeks (maximum dosing interval: none). If CST increases by ≥10% to <20% (from reference CST), the current interval will be maintained. If CST increases by ≥20% (from reference CST), the interval will be reduced by 4 weeks (minimum dosing interval: 4 weeks).

At each scheduled visit, the following assessments will be performed (a full list of assessments per visit is presented in Fig 1): review of concomitant medication/therapy; review of ocular and non-ocular adverse events (AEs); visual acuity test (to be performed before pupillary dilation); SD-OCT or swept-source OCT (after pupillary dilation) to measure CST; and OCT-angiography (after pupillary dilation) to assess retinal vascular density of the superficial, deep, and whole capillary plexuses. These assessments will also be performed during any unscheduled visits.

For visual acuity tests, corrected distance (5 m) visual acuity will be measured on a Landolt ring chart, and decimal visual acuity will be recorded. Best-corrected visual acuity (BCVA), which will be expressed in logMAR, will be converted from decimal visual acuity (d) using the following formula:

$$logMAR = \log (1/d)$$

Decimal visual acuity below 0.02 will be assessed by finger counting, hand motion, or light perception tests.

## Outcome measures

The primary efficacy endpoint is the change from baseline in BCVA at week 72. Secondary and exploratory efficacy endpoints are presented in Table 2. Outcomes will be evaluated at each time point up to week 72 for the CRVO-ME and

| | Enrollment | W0 | W4 | W8 | W12 | W16 | W20 | W24 | W28 | W32 | W36 | W40 | W44 | W48 | W52 | W56 | W60 | W64 | W68 | W72 | Unscheduled |
|---|---|---|---|---|---|---|---|---|---|---|---|---|---|---|---|---|---|---|---|---|---|
| **Visit Window (Days)** | D−14~D1 | D1 | (−7, +14) | (−7, +14) | (−7, +14) | (−7, +14) | (−7, +14) | (−7, +14) | (−7, +14) | (−7, +14) | (−7, +14) | (−7, +14) | (−7, +14) | (−7, +14) | (−7, +14) | (−7, +14) | (−7, +14) | (−7, +14) | (−7, +14) | (−7, +14) | (−7, +14) | |
| Scheduled visit | X | X | X | X | X | X | X | X | | | X[a] | | | | O | | | | | X | |
| **Enrollment** | | | | | | | | | | | | | | | | | | | | | | |
| Informed consent | X[b] | | | | | | | | | | | | | | | | | | | | | |
| Review of inclusion and exclusion criteria | X | | | | | | | | | | | | | | | | | | | | | |
| Medical and surgical history | X | | | | | | | | | | | | | | | | | | | | | |
| Patient demographics | X | | | | | | | | | | | | | | | | | | | | | |
| **Interventions** | | | | | | | | | | | | | | | | | | | | | | |
| Study treatment[c] | | ← | | | | | | | | | | | | | | | | | | → | O |
| **Assessments** | | | | | | | | | | | | | | | | | | | | | | |
| Blood pressure (systolic and diastolic) | X | X[d] | | | | | | | | | | | | | X | | | | | X | |
| Refraction test[e] | X | X[d] | | | | | | | | | | | | | | | | | | | |
| Axial length test | O | O[d] | | | | | | | | | | | | | | | | | | | |
| Review of concomitant medication/therapy | | X | X | X | X | X | X | X | X[g] | X[g] | X[g] | X[g] | X[g] | X[g] | X | X[g] | X[g] | X[g] | X[g] | X | X |
| Review of AEs and device malfunctions[f] | | ← | | | | | | | | | | | | | | | | | | → | X |
| Visual acuity test[e] | X | X[d] | X | X | X | X | X | X | X[g] | X[g] | X[g] | X[g] | X[g] | X[g] | X | X[g] | X[g] | X[g] | X[g] | X | X |
| Intraocular pressure test[e,h] | X | X[d] | X | X | X | X | X | X | X[g] | X[g] | X[g] | X[g] | X[g] | X[g] | X | X[g] | X[g] | X[g] | X[g] | X | X |
| SD-OCT or SS-OCT[e,i] | X | X[d] | X | X | X | X | X | X | X[g] | X[g] | X[g] | X[g] | X[g] | X[g] | X | X[g] | X[g] | X[g] | X[g] | X | X |
| OCT-A[e,i] | X | X[d] | X | X | X | X | X | X | X[g] | X[g] | X[g] | X[g] | X[g] | X[g] | X | X[g] | X[g] | X[g] | X[g] | X | X |
| FA[e] | X | X[d] | | | | | | | | | | | | | X | | | | | | |
| CFP[e] | X | X[d] | | | | | | | | | | | | | X | | | | | | |

X Mandatory O Optional ☐ No assessment

**Fig 1. Schedule of enrollment, visits, interventions, and assessments in RVOSTAR.** AE, adverse event; CFP, color fundus photography; D, day; FA, fluorescein angiography; IOP, intraocular pressure; OCT-A, optical coherence tomography-angiography; SD-OCT, spectral-domain optical coherence tomography; SS-OCT, swept-source optical coherence tomography; W, week. [a]This will be a scheduled visit for only patients in the observation phase. [b]Results of assessments performed as part of routine care within 14 days before day 1 (even prior to obtaining informed consent) may be used; therefore, assessments do not need to be repeated for screening. [c]The study drug on day 1 should be given within 28 days after obtaining informed consent, and the next dose should be at least 21 days apart. Finger counting will be assessed within 15 minutes after study drug administration to confirm that there is no problem with visual function. [d]Assessment not needed on day 1 if performed at screening. [e]Mandatory assessment for the study eye. [f]Information about malfunctions of mechanical or equipment components of the study drug should be entered on the electronic data capture AEs page if there is a risk of health damage, even if no AE is involved. [g]Measured during the visit. [h]IOP of the study eye will be measured before pupillary dilation for ocular tests, and if IOP before dilation is ≥ 30 mmHg, administration of dilating drops and study drug should be stopped. If possible, IOP of the study eye will also be measured 30 minutes after study drug administration. [i]Performed after pupillary dilation.

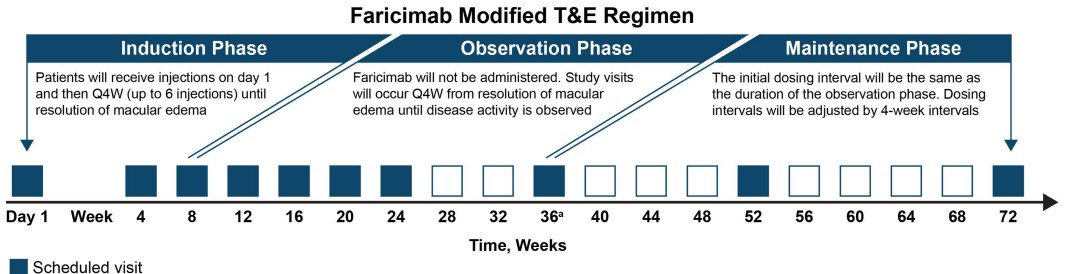

**Faricimab Modified T&E Regimen**

| Induction Phase | Observation Phase | Maintenance Phase |
|---|---|---|
| Patients will receive injections on day 1 and then Q4W (up to 6 injections) until resolution of macular edema | Faricimab will not be administered. Study visits will occur Q4W from resolution of macular edema until disease activity is observed | The initial dosing interval will be the same as the duration of the observation phase. Dosing intervals will be adjusted by 4-week intervals |

Day 1 Week 4 8 12 16 20 24 28 32 36[a] 40 44 48 52 56 60 64 68 72

**Time, Weeks**

■ Scheduled visit

**Fig 2. Faricimab modified treat-and-extend (T&E) regimen to be used in RVOSTAR.** [a]Scheduled visit at week 36 is only for patients in the observation phase.

**Table 2. Efficacy endpoints for the RVOSTAR study.**

| Efficacy endpoint | Description |
|---|---|
| Primary endpoint | • Change from baseline in BCVA at week 72 (logMAR) |
| Secondary endpoints (all time points up to week 72 unless specified) | • BCVA[a], change from baseline in BCVA (logMAR)[a], and proportion of patients with an improvement/worsening of BCVA from baseline (≥0.3 logMAR)<br>• CST and change from baseline in CST[a]<br>• Proportion of patients on each faricimab dosing interval[a]<br>• Mean total number of faricimab injections delivered[a]<br>• Number of days of the observation phase (assessed at week 72)<br>• Proportion of patients who did not receive additional doses of faricimab during the observation phase (assessed at week 72) |
| Exploratory endpoints (all time points up to week 72 unless specified) | • Proportion of patients with resolution of ME<br>• Proportion of patients with absence of IRF, absence of SRF, or both, as well as fluid volumes and their change from baseline<br>• Proportion of patients with absence of retinal ischemia; macular and total retinal areas of ischemic non-perfusion (capillary loss) and their change from baseline; and proportion of patients who transitioned from non-ischemic CRVO to ischemic CRVO<br>• Proportion of patients with absence of macular leakage, macular area microaneurysms, microaneurysm leakage and number of microaneurysms and change from baseline in vascular density of the superficial, deep, and whole capillary plexuses<br>• Relationships between baseline parameters and change from baseline in BCVA, change from baseline in CST, number of injections, dosing interval, and other efficacy parameters<br>• Relationships between variation of CST up to week 24 (SD of each patient based on CST values from weeks 4–24) and BCVA, dosing interval, and other efficacy parameters<br>• Exploration of factors affecting the dosing interval at week 72<br>• Comparison of parameters at baseline and week 72 of patients who received multiple injections vs patients who received only 1 injection of faricimab during the induction phase<br>• Number of days to achieve the best BCVA and lowest CST after initiation of faricimab<br>• Area under the curve of BCVA from day 1 to week 24<br>• Proportion of loss of peak vision (0.1, 0.2, and ≥0.3 logMAR worsening from best BCVA) and exploration of contributing demographic factors<br>• Incidence of new epiretinal membranes in patients with absence of epiretinal membranes at baseline |

BCVA, best-corrected visual acuity; CRVO, central retinal vein occlusion; CST, central subfield thickness; ETDRS, Early Treatment Diabetic Retinopathy Study; IRF, intraretinal fluid; logMAR, logarithm of the minimum angle of resolution; ME, macular edema; SD, standard deviation; SRF, subretinal fluid.

[a]Outcomes are evaluated by the presence or absence of retinal ischemia at baseline, defined as ≥10 disc areas of capillary occlusion on ETDRS 7-field fundus photography (or an equivalent range). All others are defined as non-retinal ischemia.

HRVO-ME populations. Secondary efficacy endpoints up to week 72 may be stratified by the presence or absence of retinal ischemia at baseline (defined as ≥10 disc areas of capillary occlusion on Early Treatment Diabetic Retinopathy Study 7-field fundus photography, or an equivalent range). The incidence and severity of ocular and non-ocular AEs will also be reported.

## Statistical analysis

The study aims to enroll 72 patients with CRVO-ME or HRVO-ME (≤14 patients with HRVO-ME). The rate of HRVO-ME in the COMINO trial was approximately 20%; therefore, the upper limit of HRVO-ME enrollment into RVOSTAR is 14 patients. Based on the COMINO study [9], and assuming a true value of 16.9 letters and a standard deviation (SD) of 16.45 for the change from baseline in BCVA at week 72 (primary endpoint), a sample size of 50 patients will provide ≥90% probability that the point estimate of the primary endpoint is ≤ 3 letters below the true value (one-sided calculation). In this study, considering factors such as treatment costs being borne by the patient, the dropout rate up to week 72 is predicted to be 30%. Therefore, the target is 72 patients.

BCVA will be analyzed using mixed models for repeated measures; the model will include visit (categorical variable) and baseline BCVA (continuous variable) as fixed effects, and an unstructured covariance structure will be assumed for modeling within-patient errors. For missing data, missing-at-random will be assumed. A descriptive analysis of other outcomes will be performed: means, medians, ranges, interquartile range, and SDs will be calculated for continuous variables and absolute and relative frequencies will be calculated for categorical variables. In addition, as an exploratory analysis, we will evaluate the consistency of the change from baseline in BCVA at week 72 obtained in the COMINO study [9]. No statistical hypotheses will be set, and no tests will be performed in the consistency evaluation. Data will be analyzed using SAS 9.4 (SAS Institute Inc., Cary, NC). An interim analysis will be conducted at week 24.

**Trial status and data availability**

The RVOSTAR study is currently recruiting patients, with the first patient enrolled on July 1, 2025. Last patient in is expected by December 31, 2026, and last patient out is expected by June 30, 2028. Information on participating study sites is available at the Japan Registry of Clinical Trials website (jRCTs041250001) [15]. Data from an interim analysis (at 24 weeks) will be presented at a congress in April 2028, and data from the primary analysis (at 72 weeks) will be presented at a congress in April 2029. All relevant data from this study will be made available upon study completion.

**Discussion**

The emergence of anti-VEGF agents has greatly advanced the treatment of RVO-ME [18]. However, insufficient long-term control with anti-VEGF pro re nata dosing remains a challenge in clinical practice, partially due to potential underdosing [19,20]. The Japanese RVOSTAR study protocol outlined here will enable investigation of a modified faricimab T&E regimen, which may allow for long-term maintenance of vision outcomes in patients with CRVO-ME or HRVO-ME while reducing the burden associated with frequent injections. Importantly, optimizing treatment frequency for each patient could improve real-world outcomes in clinical practice in Japan.

In the RVOSTAR protocol, the number of injections given during the induction phase, as well as the dosing intervals during the maintenance phase, can both be adjusted according to each patient's disease status (indicated by CST fluctuations), with no maximum treatment interval. Previously, in the pivotal phase 3 COMINO study, vision improvement could be maintained with faricimab treatment for 72 weeks in patients with CRVO-ME or HRVO-ME, with a median of ~4–5 injections given between 24 and 72 weeks [11]. Of note, the COMINO study used a strict T&E regimen (i.e., once the dosing interval was reduced, it was difficult to re-extend the interval) [10,11]. Notably, previous faricimab T&E regimens have capped the maximum dosing interval at Q16W [10,11], whereas in the RVOSTAR protocol, there is no maximum dosing interval. In addition, close monitoring of CST fluctuations as an indication of recurrence of ME is critical for improving disease prognosis in patients. For example, in a study evaluating foveal thickness fluctuations in recurrent CRVO-ME, patients with larger fluctuations during the observation period (no anti-VEGF treatment) had worse visual acuity with anti-VEGF pro re nata treatment by the end of the study [21]. Overall, the RVOSTAR study could reduce unnecessary overdosing and lower the burden on patients and the healthcare system, including with regard to factors known to impact anti-VEGF treatment adherence, such as drug costs, scheduling requirements, and logistics of scheduled visits [14,22].

The RVOSTAR study has a robust design (unmasked, single-arm, multicenter, prospective interventional) based on the COMINO study [10,11] and will recruit Japanese patients with CRVO-ME or HRVO-ME. The number of patients with HRVO-ME recruited will be limited to allow CRVO-ME to be the focus of the study. Study insights will be particularly important for patients with CRVO-ME, in which long-term maintenance of vision has been a challenge in previous trials [3–5]. Although the COMINO study did not define retinal ischemic type or conduct subgroup analysis separating ischemic and non-ischemic types [10], RVOSTAR will define retinal ischemic type in advance and will perform subgroup analysis by retinal ischemic and non-ischemic types. Importantly, this study design allows for recurrence of disease activity to guide treatment intervals, allowing for personalized treatment.

Previous clinical trials have tended to report a higher number of injections compared with what is usually observed in real-world clinical practice [23]; hence, this is a potential limitation of the RVOSTAR study. Nevertheless, the insights from RVOSTAR will help to optimize the number of injections required to maintain long-term efficacy in real-world clinical practice.

In conclusion, the modified faricimab T&E regimen being evaluated in the RVOSTAR study will provide important information on the efficacy and safety of long-term treatment of Japanese patients with CRVO-ME or HRVO-ME. The results from RVOSTAR may help address an unmet need for optimizing treatment outcomes while reducing treatment burden.

## Supporting information

**S1 Table. Full exclusion criteria.**
(PDF)

**S1 Fig. RVOSTAR dosing regimen.**
(PDF)

**S1 File. Protocol EN.** RVOSTAR study protocol (English).
(PDF)

**S2 File. Protocol JP.** RVOSTAR study protocol (Japanese).
(PDF)

**S1 Checklist. SPIRIT checklist.**
(PDF)

## Author contributions

**Conceptualization:** Mineo Kondo, Masahiko Shimura, Akitaka Tsujikawa, Yuki Muraoka, Takahiro Kogo, Yuki Akiyama, Yuki Hama, Jun Tsujimura, Keisuke Iwasaki, Motohiro Kamei.

**Methodology:** Mineo Kondo, Masahiko Shimura, Akitaka Tsujikawa, Yuki Muraoka, Takahiro Kogo, Yuki Akiyama, Yuki Hama, Jun Tsujimura, Keisuke Iwasaki, Motohiro Kamei.

**Project administration:** Jun Tsujimura, Keisuke Iwasaki.

**Writing – original draft:** Mineo Kondo, Masahiko Shimura, Akitaka Tsujikawa, Yuki Muraoka, Takahiro Kogo, Yuki Akiyama, Yuki Hama, Jun Tsujimura, Keisuke Iwasaki, Motohiro Kamei.

**Writing – review & editing:** Mineo Kondo, Masahiko Shimura, Akitaka Tsujikawa, Yuki Muraoka, Takahiro Kogo, Yuki Akiyama, Yuki Hama, Jun Tsujimura, Keisuke Iwasaki, Motohiro Kamei.

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
