## [Decision Letter · Decision Letter 0]

7 Oct 2025

A multicenter, single-arm study using a modified faricimab treat-and-extend regimen in patients with macular edema due to central retinal vein occlusion: RVOSTAR study design protocol

PONE-D-25-46857

Dear Dr. Kondo,

We’re pleased to inform you that your manuscript has been judged scientifically suitable for publication and will be formally accepted for publication once it meets all outstanding technical requirements.

Kind regards,

Shinji Kakihara, M.D.,Ph.D.

Academic Editor

PLOS ONE

Journal Requirements:

“This study was funded by Chugai Pharmaceutical Co., Ltd., a member of the Roche Group. Chugai Pharmaceutical Co., Ltd. was involved in the study design and preparation of the manuscript. Medical writing assistance was provided by Nilisha Fernando, PhD, of Envision Pharma Group, and funded by Chugai Pharmaceutical Co., Ltd. Envision Pharma Group’s services complied with international guidelines for Good Publication Practice (GPP 2022).”

Please respond by return e-mail so that we can amend your financial disclosure and competing interests on your behalf.

“I have read the journal's policy and the authors of this manuscript have the following competing interests: M.Kondo: financial support from Alcon Japan, Chugai, HOYA, Santen, and Senju; consulting fees from Chugai; royalties from Alcon Japan, Bayer, HOYA, Kowa, Novartis, Otsuka, Santen, and Senju. M.S.: financial support from Bayer, Chugai, Kowa, Novartis, Otsuka, Roche, Santen, and Senju; consulting fees from Bayer, Boehringer Ingelheim, Chugai, Cimic, and Nikki HD. AT: financial support from Novartis, Santen, and Senju; consulting fees from Santen and Senju; renumeration from Bayer, Chugai, Novartis, Santen, and Senju. Y.M.: financial support from Alcon Japan, AMO Japan, Bayer Yakuhin, Canon, Chugai, HOYA, Johnson & Johnson K.K., Novartis, ROHTO, Santen, and Senju. T.K., Y.A., Y.H.: none. J.T., K.I.: employees of Chugai Pharmaceutical Co., Ltd. M.Kamei: financial support from Alcon Japan, Chugai, HOYA, Kowa, Menicon, Otsuka, Santen, Senju, Sony, and Wakamoto; royalties from Alcon Japan, Bayer, Chugai, Daicel, HOYA, Kowa, Novartis, Santen, Senju, Sony, and Wakamoto.”

We note that you received funding from a commercial source: Chugai Pharmaceutical Co., Ltd. M.Kamei

Please respond by return email with your amended Competing Interests Statement and we will change the online submission form on your behalf.

Additional Editor Comments (optional):

This protocol was reviewed by two reviewers, and both found the study design clear and well organized. I agree with their assessment and see no need for revision. I recommend acceptance as submitted. Thanks

Reviewers' comments:

Reviewer's Responses to Questions

**Comments to the Author**

1. Does the manuscript provide a valid rationale for the proposed study, with clearly identified and justified research questions?

Reviewer #1: Yes

Reviewer #2: Yes

2. Is the protocol technically sound and planned in a manner that will lead to a meaningful outcome and allow testing the stated hypotheses?

Reviewer #1: Yes

Reviewer #2: Yes

3. Is the methodology feasible and described in sufficient detail to allow the work to be replicable?

Reviewer #1: Yes

Reviewer #2: Yes

4. Have the authors described where all data underlying the findings will be made available when the study is complete?

Reviewer #1: Yes

Reviewer #2: Yes

5. Is the manuscript presented in an intelligible fashion and written in standard English?

Reviewer #1: Yes

Reviewer #2: Yes

You may also provide optional suggestions and comments to authors that they might find helpful in planning their study.

Reviewer #1: This manuscript presents the protocol of a prospective multicenter study (RVOSTAR) evaluating the efficacy of a modified treat-and-extend regimen of faricimab for RVO-ME. The Introduction and Discussion sections provide a clear and well-structured overview of the disease background and the limitations of existing treatments, highlighting the rationale and significance of the study. The Methods section describes the study design, eligibility criteria, and outcome measures in sufficient detail, providing all the essential information expected in a protocol paper.

I believe that no specific revisions are needed, and the manuscript will be very clear and valuable for the readers.

Reviewer #2: This study investigates effective treatments for CRVO-ME. The background and objectives are clearly stated, and the significance of the research is well conveyed.

The structure is highly logical.

The discussion is thorough, and connections to prior research are well demonstrated.

I recommend acceptance of this paper.

**Do you want your identity to be public for this peer review?** For information about this choice, including consent withdrawal, please see our Privacy Policy

Reviewer #1: No

Reviewer #2: **Yes: ** Eimi Suzuki

---

## [Editor Report · Acceptance letter]

PONE-D-25-46857

PLOS ONE

Dear Dr. Kondo,

I'm pleased to inform you that your manuscript has been deemed suitable for publication in PLOS ONE. Congratulations! Your manuscript is now being handed over to our production team.

Kind regards,

on behalf of

Dr. Shinji Kakihara

Academic Editor

PLOS ONE